# Attention-deficit hyperactivity disorder symptoms and brain morphology: Examining confounding bias

**Lorenza Dall'Aglio[1,2†], Hannah H Kim[3†], Sander Lamballais[4†], Jeremy Labrecque[5], Ryan L Muetzel[1‡], Henning Tiemeier[1,3*‡]**

[1]Department of Child and Adolescent Psychiatry, Erasmus MC University Medical Center Rotterdam-Sophia Children's Hospital, Rotterdam, Netherlands; [2]The Generation R Study Group, Erasmus MC University Medical Center Rotterdam, Rotterdam, Netherlands; [3]Department of Social and Behavioral Sciences, Harvard TH Chan School of Public Health, Boston, United States; [4]Department of Clinical Genetics, Erasmus MC, Rotterdam, Netherlands; [5]Department of Epidemiology, Erasmus MC, Rotterdam, Netherlands

**\*For correspondence:**
tiemeier@hsph.harvard.edu

[†]These authors contributed equally to this work

[‡]Ryan L. Muetzel and Henning Tiemeier are co-last authors.

**Competing interest:** The authors declare that no competing interests exist.

## Abstract

**Background:** Associations between attention-deficit/hyperactivity disorder (ADHD) and brain morphology have been reported, although with several inconsistencies. These may partly stem from confounding bias, which could distort associations and limit generalizability. We examined how associations between brain morphology and ADHD symptoms change with adjustments for potential confounders typically overlooked in the literature (aim 1), and for the intelligence quotient (IQ) and head motion, which are generally corrected for but play ambiguous roles (aim 2).

**Methods:** Participants were 10-year-old children from the Adolescent Brain Cognitive Development (*N* = 7722) and Generation R (*N* = 2531) Studies. Cortical area, volume, and thickness were measured with MRI and ADHD symptoms with the Child Behavior Checklist. Surface-based cross-sectional analyses were run.

**Results:** ADHD symptoms related to widespread cortical regions when solely adjusting for demographic factors. Additional adjustments for socioeconomic and maternal behavioral confounders (aim 1) generally attenuated associations, as cluster sizes halved and effect sizes substantially reduced. Cluster sizes further changed when including IQ and head motion (aim 2), however, we argue that adjustments might have introduced bias.

**Conclusions:** Careful confounder selection and control can help identify more robust and specific regions of associations for ADHD symptoms, across two cohorts. We provided guidance to minimizing confounding bias in psychiatric neuroimaging.

**Funding:** Authors are supported by an NWO-VICI grant (NWO-ZonMW: 016.VICI.170.200 to HT) for HT, LDA, SL, and the Sophia Foundation S18-20, and Erasmus University and Erasmus MC Fellowship for RLM.

## Editor's evaluation

This study provides important and useful information to researchers in brain morphology and ADHD. The strength of the evidence presented is convincing and solid.

## Introduction

Large strides have been made in the identification of neuroanatomical correlates of psychiatric problems, with attention-deficit/hyperactivity disorder (ADHD) being a prominent example. ADHD is the most prevalent neurodevelopmental disorder in children worldwide and is characterized by atypical levels of inattention, hyperactivity, and/or impulsivity (*American Psychiatric Association, 2013*). Structural magnetic resonance imaging studies have highlighted that children with ADHD show widespread morphological differences, such as in the basal ganglia (*Nakao et al., 2011*), subcortical areas (*Hoogman et al., 2017*), and frontal, cingulate, and temporal cortices, compared to children without the disorder (*Hoogman et al., 2019*; *Shaw et al., 2013*).

Consistently identifying the neuroanatomical substrate of ADHD, however, remains challenging. A recent meta-analysis did not find convergence across the literature on brain differences in children and adolescents with ADHD (*Samea et al., 2019*). One possible explanation for this inconsistency is the multifaceted nature of ADHD, in which children with the disorder have heterogeneous presentations on several cognitive and emotional domains, which could stem from distinct brain structural substrates. Other explanations regard study design. If suboptimal, it may lead to biased estimates and lack of generalizability, thus potentially concealing robust and replicable relations of brain morphology with ADHD. The present study focuses on confounding, a common source of bias in etiological studies.

Confounding bias arises when a third variable affects both the determinant (independent variable) and outcome (dependent variable) of interest (i.e., is a common cause) (*VanderWeele, 2019*). Confounding leads to over- or underestimation of the true effect between determinant and outcome and can even change the direction of an association. To minimize confounding bias, appropriate confounder control is paramount, although it is challenging, especially in observational studies like most neuroimaging studies of ADHD. Previous literature and expert knowledge can guide the identification of potential confounders (*Hernan and Robins, 2020*), which can then be appropriately adjusted for in regression models or using methods such as restriction, standardization, or propensity scores.

Within neuroimaging studies of ADHD, except for a few large investigations (*Hoogman et al., 2017*; *Mous et al., 2014*; *Bernanke et al., 2022*), studies have generally matched or adjusted for a few demographic variables (e.g., age and sex) and neuroimaging metrics or parameters. Of the 19 studies included in a systematic review of neuroimaging studies on ADHD (*Saad et al., 2020*), 17 adjusted or matched for age in their analyses, 14 for sex, 9 for neuroimaging-related variables like head motion during scanning, and 8 for the intelligence quotient (IQ) (*Supplementary file 1a*). Further potential confounders should, however, be considered. For instance, socioeconomic status (SES) is related to both higher risk for ADHD and variation in cortical brain structure (*Russell et al., 2016*; *Noble et al., 2015*). Thus, it is likely a confounder. Lack of adjustment for SES may have therefore concealed key relations between ADHD and brain structure. Adjustment choices are dependent on the availability of large samples with data on a wide variety of covariates, which has to date been limited for psychiatric neuroimaging studies. Yet, this is rapidly changing with the advent of population neuroscience, which entails large-scale studies with neurobiological data. This lends new opportunities for further confounder adjustments to be considered in neuroimaging studies of ADHD. Conversely, previous studies have adjusted for IQ and head motion, which may not be confounders in the association between ADHD symptoms and the brain, and may thus have led to further bias in the results (*Dennis et al., 2009*).

In this study, we examined the association between brain structure and ADHD symptoms and how the selection and control for potential confounders may affect results (aim 1). Moreover, we discussed the complex role of IQ and head motion in brain structure–ADHD associations and the potential consequences of adjusting for them (aim 2). We leveraged two large, population-based cohorts: the Adolescent Brain Cognitive Development (ABCD) and the Generation R Studies. In line with most neuroimaging studies, we adopted a cross-sectional design.

## Results

### Associations between ADHD symptoms and brain morphology are widespread

We analyzed data from 10-year-old children from the ABCD (*N* = 7722, multisite) and Generation R (*N* = 2531, single-site) Studies (*Supplementary file 1b*). ADHD symptoms were measured with the Child

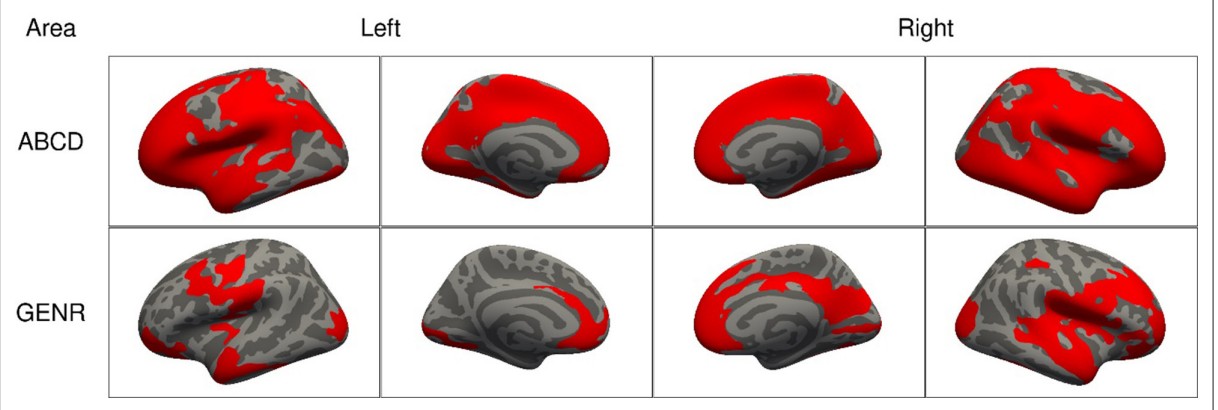

**Figure 1.** Significant clusters in the association of attention-deficit/hyperactivity disorder (ADHD) symptoms with cortical surface area based on the Adolescent Brain Cognitive Development (ABCD) and Generation R Studies, for model 1. *Note.* Rows represent the results for the ABCD or Generation R Studies, and the columns represent the left and right hemispheres. Regions in red represent significant clusters from model 1 (adjusted for sex, age, race/ethnicity, and site [ABCD only]).

The online version of this article includes the following figure supplement(s) for figure 1:

**Figure supplement 1.** Significant clusters in the association of attention-deficit/hyperactivity disorder (ADHD) symptoms with cortical volume (top) and thickness (bottom) based on the Adolescent Brain Cognitive Development (ABCD) and Generation R Studies, for model 1.

**Figure supplement 2.** Significant clusters in the association of attention-deficit/hyperactivity disorder (ADHD) diagnosis with cortical surface area (top), volume (middle), and thickness (bottom) for the Adolescent Brain Cognitive Development (ABCD) Study, for model 1.

Behavioral Checklist (CBCL). $T_1$-weighted images were obtained with 3T scanners (*Casey et al., 2018*; *Kooijman et al., 2016*). We ran vertex-wise linear regression models for ADHD with cortical surface area, volume, and thickness. Results for surface area, which constituted the main findings here, are presented in-text, while findings for volume and thickness in the figure supplements. We adjusted for demographic and study characteristics which have been generally considered by previous literature (*Supplementary file 1a*): age, sex, ethnicity, and study site (ABCD only). We refer to this model as model 1, as further adjustments for confounders are outlined in subsequent steps.

We found that higher ADHD symptoms were associated with less bilateral surface area in both cohorts. As shown in *Figure 1*, associations were widespread, as the clusters of association covered 1165.7 cm$^2$ of the cerebral cortex in the ABCD Study, and 446.1 cm$^2$ in the Generation R Study. Across both cohorts, we consistently identified clusters for surface area in the lateral occipital, postcentral, rostral middle and superior frontal, and superior parietal cortices. For cortical thickness, we found two small frontal clusters in the ABCD Study (16.1 cm$^2$) and no clusters in the Generation R Study, which suggests that cortical thickness does not relate or does not relate strongly to ADHD, in line with prior literature (*Hoogman et al., 2019*; *Figure 1—figure supplement 1*).

## Confounder selection: socioeconomic and maternal behavioral factors

Next, we considered factors that have been previously linked to ADHD and brain structure in the literature, and are thus potential confounders. To illustrate this background knowledge and the assumptions about relations between variables, we used Directed Acyclic Graphs (DAGs), a type of causal diagram (*Hernan and Robins, 2020*). These guide the identification (and dismissal) of covariates that may act as confounders. Of note, while assumptions may not hold, this theoretical approach is preferred to methods selecting confounders based on model statistics (*Lee, 2014*). The DAGs are depicted in *Figure 2* and *Figure 2—figure supplements 1 and 2*, and the rationales for variable inclusion are explained below and in the *Methods*.

Based on the literature, lower SES is associated with a higher risk for ADHD (*Russell et al., 2016*) and with variation in cortical brain structure (*Noble et al., 2015*). Thus, confounding by socioeconomic factors in the relation between ADHD and brain morphology is likely. We therefore additionally adjusted for a second set of confounders (model 2) related to SES: household income, maternal education, and maternal age at childbirth.

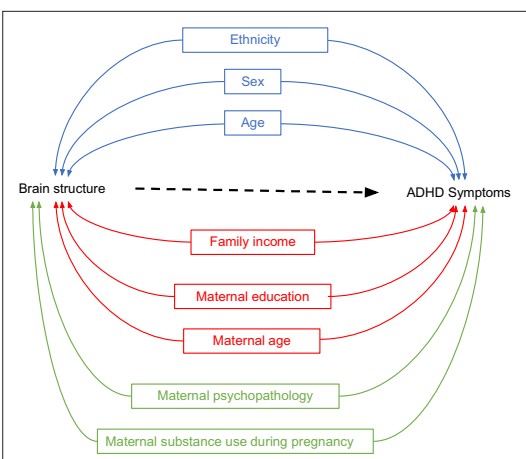

**Figure 2.** Directed Acyclic Graphs (DAGs) for brain structure and attention-deficit/hyperactivity disorder (ADHD) symptoms (simplified). *Note.* DAGs illustrating potential confounders in the association between brain structure and ADHD symptoms for three *sequential* models. Model 1 included demographic and study characteristics: sex, age, ethnicity, and study site (Adolescent Brain Cognitive Development [ABCD] only) (in blue). Model 2 additionally included socioeconomic status factors: family income, maternal education, and maternal age at childbirth (in red). Model 3 additionally incorporated postnatal maternal psychopathology and maternal substance use during pregnancy (in green).

The online version of this article includes the following figure supplement(s) for figure 2:

**Figure supplement 1.** Illustration of Directed Acyclic Graphs (DAGs) using family income as an example.

**Figure supplement 2.** Directed Acyclic Graphs (DAGs) for brain structure and attention-deficit/hyperactivity disorder (ADHD) symptoms (complete).

Moreover, several factors concerning maternal behavior, pre- and postnatally, have been associated with both ADHD and brain morphology. For instance, prenatal exposure to substances is known to increase the risk of developing ADHD symptoms and has been associated with variation in cerebral volume and surface area (*Eilertsen et al., 2017*; *Lees et al., 2020*). Postnatal maternal psychopathology has been linked to higher child ADHD symptoms (*Clavarino et al., 2010*) and smaller brain volume in children (*Zou et al., 2019*). Thus, in model 3 we additionally adjusted for prenatal exposure to substance use (tobacco and cannabis), and postnatal maternal psychopathology.

## Adjusting for additional confounders led to reductions in the clusters of association

Adjustments for SES (model 2) led to reductions in the spatial extent of the clusters for surface area and volume in both cohorts (*Figure 3*). For surface area, cluster sizes for ADHD symptoms reduced from 1165.7 cm$^2$ in model 1 to 952.8 cm$^2$ in model 2 (=−18%) in the ABCD Study, and from 446.1 to 229.6 cm$^2$ (=−49%) in the Generation R Study. Similar reductions were observed for volume and thickness (*Figure 3—figure supplement 1*). After adjusting for the confounders added in model 3, across both cohorts, we consistently identified clusters for surface area in the cuneus, precuneus, fusiform, inferior parietal, isthmus of the cingulate, pericalcarine, pre- and postcentral, rostral middle and superior frontal, superior temporal and supramarginal cortices.

## Similar results were observed for ADHD diagnosis

To explore whether the results observed for associations between brain morphology and ADHD symptoms applied to children with an ADHD diagnosis, we repeated the primary analysis using the ADHD diagnostic data from the Kiddie Schedule for Affective Disorders and Schizophrenia (KSADS) in the ABCD Study. In line with our primary results, ADHD diagnosis was associated with less bilateral surface area and volume. Compared to clusters for ADHD symptoms, those associated with ADHD diagnosis were smaller, but overlapping (*Figure 1—figure supplement 2*). We observed similar patterns of reduction in the spatial extent of the clusters after adjusting for each set of confounders (*Figure 3—figure supplement 2*). For surface area, cluster sizes for ADHD symptoms covered 234.4 cm$^2$ in model 1 and reduced to 199.5 cm$^2$ in model 2 (=−15%), and 55.5 cm$^2$ in model 3 (=−72%, compared to model 2).

## Beta coefficients generally decreased after confounder adjustments, but may also increase

Surface-based studies generally focus on the spatial extent of cortical clusters associated with the phenotype, but, in this study, we also explored how confounding adjustments affected the regression coefficients for ADHD symptoms (*Figure 4*).

At a vertex-wise level, adjusting for socioeconomic and maternal factors (model 3) led to reductions in the beta coefficients, across the brain, for both cohorts (*Figure 4—figure supplement 1*).

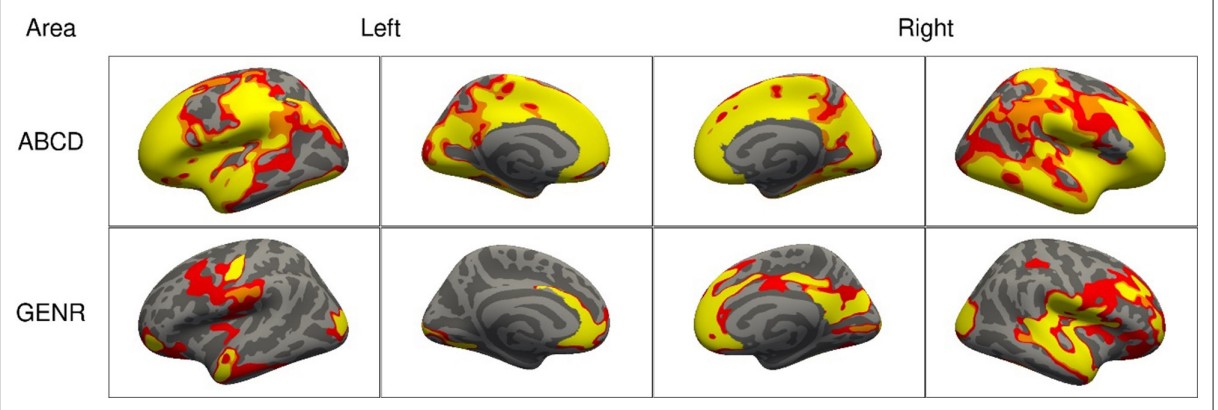

**Figure 3.** Significant clusters in the association of attention-deficit/hyperactivity disorder (ADHD) symptoms with cortical surface area based on the Adolescent Brain Cognitive Development (ABCD) and Generation R Studies, for models 1–3. *Note.* Rows represent the results for the ABCD or Generation R Studies, and the columns represent the left and right hemispheres. The colors denote the different models. Regions in red represent significant clusters from model 1 (sex, age, race/ethnicity, and site [ABCD only]), orange from model 2 (model 1 + family income, maternal education, and maternal age at childbirth), and yellow from model 3 (model 2 + maternal smoking, substance use during pregnancy, psychopathology).

The online version of this article includes the following figure supplement(s) for figure 3:

**Figure supplement 1.** Significant clusters in the association of attention-deficit/hyperactivity disorder (ADHD) symptoms with cortical volume (top) and thickness (bottom) based on the Adolescent Brain Cognitive Development (ABCD) and Generation R Studies, for models 1–3.

**Figure supplement 2.** Significant clusters in the association of attention-deficit/hyperactivity disorder (ADHD) diagnosis with cortical surface area (top), volume (middle), and thickness (bottom) for the Adolescent Brain Cognitive Development (ABCD) Study, for models 1–3.

Of note, some beta coefficients also showed increases. As confounding bias may lead to under- or overestimation, it is not surprising to observe both decreases and increases in the average beta coefficients after adjustments.

At an anatomical region level, where estimates of vertices within a given Desikan–Killiany region were averaged, beta coefficients for surface area tended to decrease from model 1 to 2 by approximately 15% (*Figure 4*, *Figure 4—figure supplement 2*). Further adjustments from model 2 to 3 led to decreases in the average beta coefficients of certain regions and increases in others. Similar patterns were found for volume (*Figure 4—figure supplements 3 and 4*). The average beta coefficients per region correlated moderately to strongly between the ABCD and Generation R Studies for surface area (Spearman $r_{M1}$ = 0.84, $r_{M2}$ = 0.83, $r_{M3}$ = 0.83) and volume (Spearman $r_{M1}$ = 0.57, $r_{M2}$ = 0.57, $r_{M3}$ = 0.70) (*Figure 4—figure supplement 5*).

## IQ may be a confounder, mediator, or collider in neuroanatomical studies of ADHD

We considered one additional scenario which included IQ, a factor that is often adjusted for in previous studies (*Supplementary file 1a*). However, based on prior literature, it holds an ambiguous role in structural anatomy–ADHD relations. Previous studies found that children with ADHD scored lower on IQ than children without ADHD (*Bridgett and Walker, 2006*). Differential brain structure with levels of IQ has also been shown (*Mcdaniel, 2005*). However, the directions of causation between these variables remain unclear (*Gallo and Posner, 2016*). IQ may therefore be a confounder, collider, and/ or mediator in the relation between brain structure and ADHD, as depicted in the DAGs in *Figure 5* and *Figure 5—figure supplement 1*.

First, it could be argued that IQ is partly innate and precedes brain development and ADHD, making it a confounder (*Figure 5A*). Second, IQ may lie in the pathway between brain structure and ADHD and therefore act as a mediator (*Figure 5B*). It is conceivable that cognitive differences, as a consequence of subtle neurodevelopmental differences (*Lee et al., 2019*), could underlie ADHD. Adjusting for a mediator would lead to bias when estimating the total association between brain structure and ADHD (*VanderWeele, 2016*). Third, brain structure may impact intelligence scores (*Lee et al., 2019*), and ADHD symptoms may affect IQ test performance (*Jepsen et al., 2009*; *Figure 5C*). A variable that is independently caused by the outcome and the determinant is also known as a

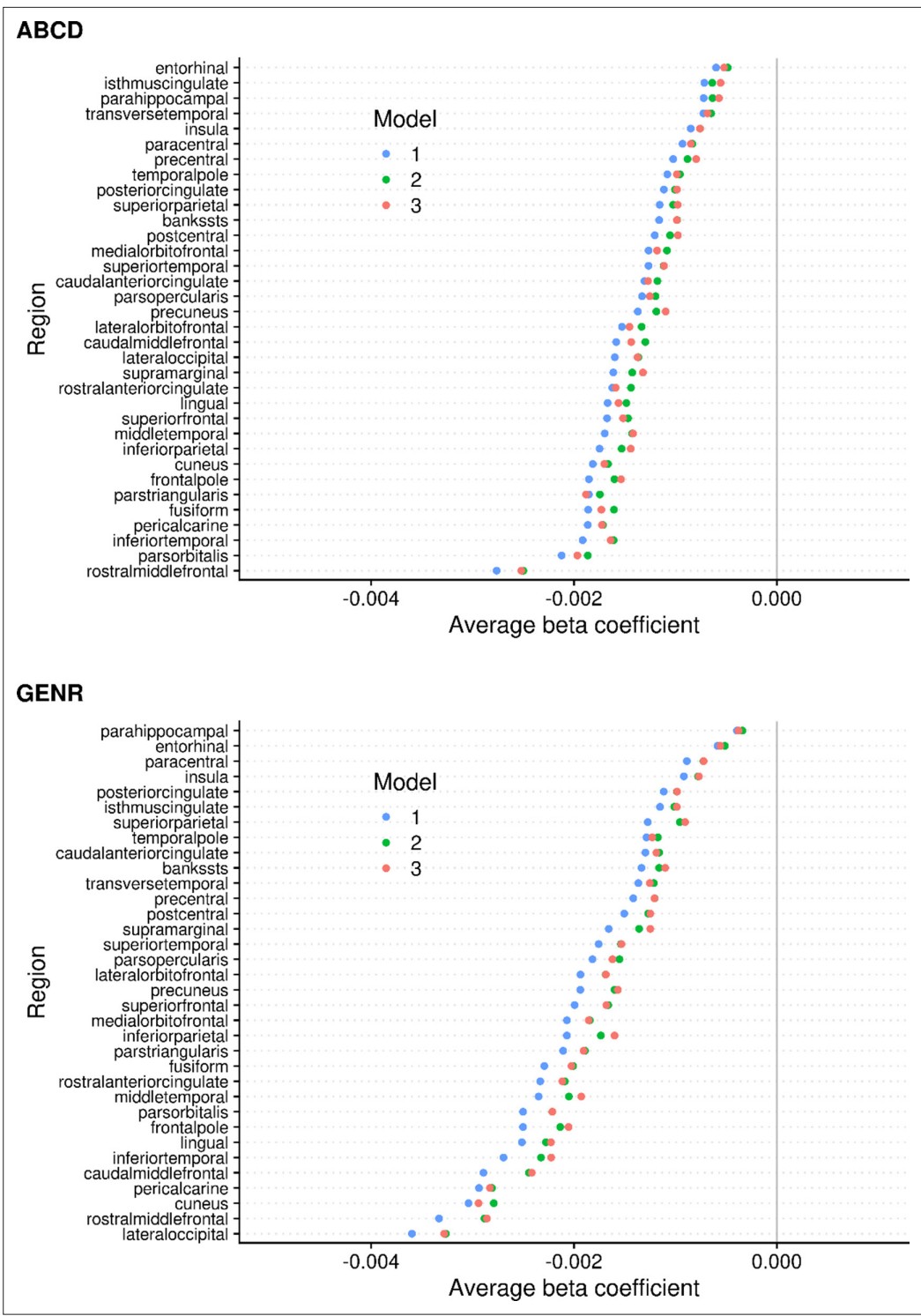

**Figure 4.** Region-based average regression coefficients for surface area in the Adolescent Brain Cognitive Development (ABCD) and Generation R Studies. *Note.* The colors denote the different models, and the circles denote the average of all the betas within that region. The regions are based on the Desikan–Killiany atlas. Results for the ABCD and Generation R Studies are, respectively, shown on the top and bottom.

The online version of this article includes the following figure supplement(s) for figure 4:

**Figure supplement 1.** The vertex-wise change in effect sizes between models 1 and 3 for cortical surface area (top), volume (middle), and thickness (bottom).

*Figure 4 continued on next page*

*Figure 4 continued*

**Figure supplement 2.** Change in the regional average betas for surface area in the Adolescent Brain Cognitive Development (ABCD) Study (**A, B**) and Generation R Study (**C, D**).

**Figure supplement 3.** Region-based average regression coefficients for cortical volume in the Adolescent Brain Cognitive Development (ABCD) Study (top) and Generation R (bottom).

**Figure supplement 4.** Change in the regional average betas for volume in the Adolescent Brain Cognitive Development (ABCD) Study (**A, B**) and Generation R Study (**C, D**).

**Figure supplement 5.** Scatterplot of Spearman correlations between region-based average regression coefficients from the Adolescent Brain Cognitive Development (ABCD) Study (*x*-axis) and the Generation R Study (*y*-axis).

collider, and adjusting for it leads to (collider) bias. Here, we explored the impact of adjusting for IQ when examining the relation between brain morphology and ADHD (model 4).

## Adjustments for IQ led to further cluster reductions

After additionally adjusting for IQ, the spatial extent of the clusters associated with ADHD symptoms reduced further in both cohorts (*Figure 6*). For surface area, compared to model 3, clusters reduced from 760.2 to 605.1 cm$^2$ (=−20%) for the ABCD Study, and from 208.6 to 93.1 cm$^2$ for the Generation R Study (=−55%). Clusters of association for surface area in model 4 were located in the fusiform, inferior parietal, insula, lateral occipital, middle temporal, pericalcarine, pre- and postcentral, precuneus, rostral middle, and superior frontal, superior parietal and temporal, and supramarginal cortices. Findings for volume and thickness are shown in *Figure 6—figure supplement 1*.

## Head motion does not induce confounding bias, but information bias

A final scenario was also included, to reflect the commonly used adjustments for head motion during scanning (*Supplementary file 1a*). Motion can be a large source of bias in neuroimaging studies which is important to address. While it does not meet the criteria for confounding as it is not a common cause of ADHD problems and brain morphology (*Hernan and Robins, 2020*), head motion can induce measurement error of brain morphology (*Van de Walle et al., 1997*; *Figure 7*). This is also referred to as information bias and can distort estimates from their true value.

The amount of measurement error in brain morphology may differ across children with versus without ADHD. In fact, children with impulsivity and inattention have been shown to move more during MRI scanning (*Thomson et al., 2021*; *Kong et al., 2014*), determining different levels of error in the brain morphology assessments (*Figure 7*, path from ADHD symptoms to motion to error in MRI measurement). In this scenario, adjusting for motion might lead to two situations. On one hand, since motion is a consequence of the outcome (ADHD), adjustments would lead to bias (*Westreich, 2012*). On the other hand, not adjusting for motion would also lead to bias because part of the observed relation between ADHD symptoms and brain structure would be due to the higher head motion (and thus the underestimation of the cortical values) of children with ADHD. In this study, we explored the effect of adjusting for motion during scanning in the relation between brain morphology and ADHD (model 5).

## Adjustments for head motion led to increases in clusters

After additional adjustments for head motion, the spatial extent of the clusters generally increased. For surface area, compared to model 3, clusters increased from 760.2 to 936.4 cm$^2$ (=+23.2%) for the ABCD Study and from 208.6 to 239.7 cm$^2$ (=+14.9%) for the Generation R Study (*Figure 8*). Clusters of associations consistently found across cohorts were highly similar to the ones identified in model 3. Results for cortical volume and thickness are shown in *Figure 8—figure supplement 1*.

## Discussion

By leveraging two large population-based studies and adopting a literature- and DAG-informed approach to address confounding, we showed that (1) associations between brain structure and ADHD symptoms, which were initially widespread, reduced when adjusting for socioeconomic and

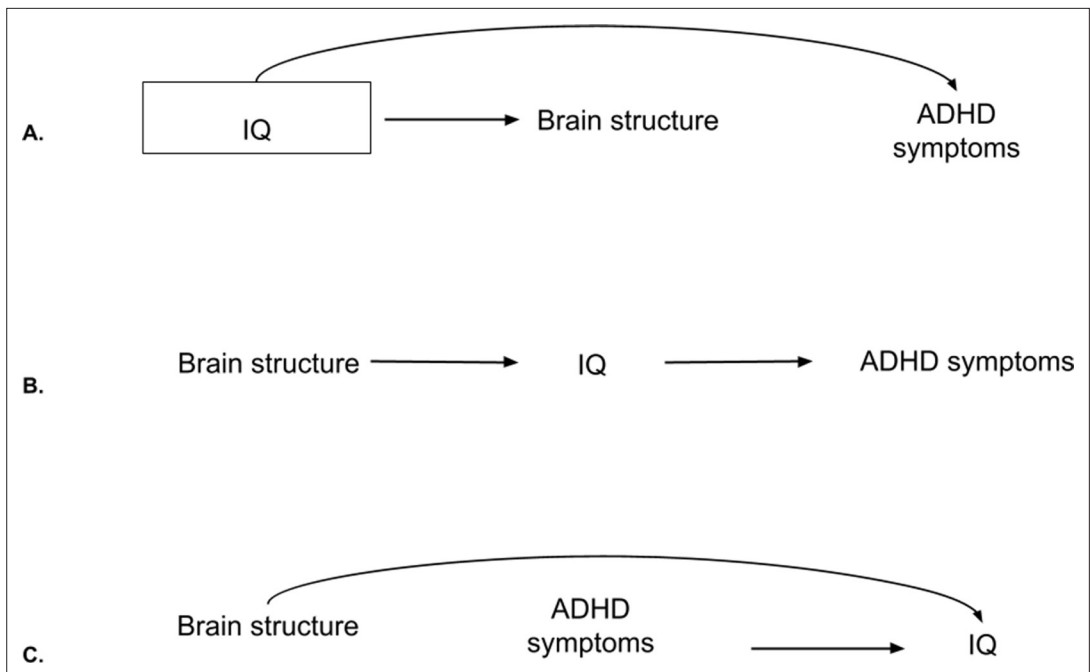

**Figure 5.** Directed Acyclic Graphs (DAGs) for intelligence quotient (IQ), brain structure, and attention-deficit/hyperactivity disorder (ADHD) symptoms. *Note.* (**A**) DAG for IQ as a confounder. In this case, adjustments are needed as the backdoor path from brain structure to ADHD symptoms through IQ is open. By adjusting (box around IQ), the path gets closed. (**B**) DAG for IQ as a mediator. Adjustments are not needed to estimate the total effect of brain structure on ADHD symptoms. (**C**) DAG for IQ as a collider. The backdoor path through IQ is already closed. Adjustments would open the path and lead to collider bias.

The online version of this article includes the following figure supplement(s) for figure 5:

**Figure supplement 1.** Directed Acyclic Graphs (DAGs) representing intelligence quotient (IQ) as a confounder, mediator, or a collider in the relation between brain structure and attention problems.

maternal behavioral confounders, and that (2) careful considerations are needed when including IQ and/or head motion due to their complex relation with ADHD and brain morphology.

## Adjustments for confounders highlighted key regions of association, observed across two large cohorts

Widespread associations between surface area and ADHD symptoms were initially identified, with higher symptoms relating to smaller brain structures, in line with previous research (*Hoogman et al., 2019*; *Gehricke et al., 2017*). After adjustments for potential confounders typically overlooked by prior literature (socioeconomic and maternal behavioral factors), approximately half of the associations remained, and considerable effect size changes were observed in both the ABCD and Generation R Studies and for all cortical measures. We observed similar patterns of cluster reductions for ADHD diagnosis in the ABCD Study.

Regions that remained associated after adjustments and which were consistently identified across cohorts were the precuneus, isthmus of the cingulate, supramarginal, pre- and postcentral, and inferior parietal cortices for both area and volume. Most of these regions (e.g., supramarginal) have been previously implicated in ADHD in clinical samples (*Saad et al., 2017*; *Lei et al., 2014*; *Solanto et al., 2009*). However, many different brain areas have been detected in association with the disorder (*Saad et al., 2020*), which may have hampered prior meta-analytic efforts to identify consistent neuroanatomical correlates for ADHD.

Of note, some inconsistencies between the ABCD and Generation R Studies, both in size of the clusters and the exact location, were observed. While we used the same processing pipelines and similar quality control procedures and measures across cohorts, potential reasons for discrepancies in results must be discussed. First, the larger sample size of the ABCD Study allows for greater power

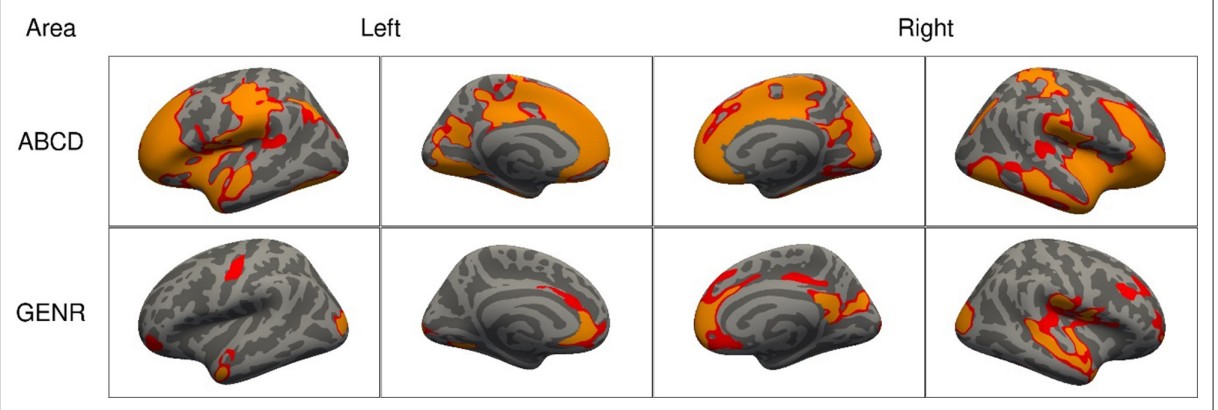

**Figure 6.** Significant clusters in the association of attention-deficit/hyperactivity disorder (ADHD) symptoms with cortical surface area based on the Adolescent Brain Cognitive Development (ABCD) and Generation R Studies, after additional adjustment for intelligence quotient (IQ). *Note.* Rows represent the results for the ABCD or Generation R Studies, and the columns represent the left and right hemispheres. The colors denote the different models, with red vertices being significant only in model 3, orange ones in both model 3 and after adjustment for IQ, and yellow ones only after adjusting for IQ.

The online version of this article includes the following figure supplement(s) for figure 6:

**Figure supplement 1.** Significant clusters in the association of attention-deficit/hyperactivity disorder (ADHD) symptoms with cortical volume (top) and thickness (bottom) based on the Adolescent Brain Cognitive Development (ABCD) and Generation R Studies, after additional adjustment for intelligence quotient (IQ).

to detect smaller effects, which led to larger associated areas. Second, the multisite structure of the ABCD Study may have introduced noise in the results (e.g., by different scanners, demographic differences), and determined the identification of associations which are not replicable in the Generation R Study. Third, the two studies include children from different populations. While both are very diverse samples, the ABCD Study is comprised of a more heterogeneous sample from the US population, which, for instance, is characterized by a wider variety of ethnicities and cultures, potentially permitting the discovery of more associations. Nevertheless, there was considerable overlap in the findings from the ABCD and Generation R Studies, with consistencies across cohorts indicating the most robust and generalizable associations.

Here, we discerned associated areas likely subject to confounding bias from areas robust to socio-economic and maternal behavioral factors, and replicable across two large cohorts. Comparisons with prior findings should be made with caution due to differences in study design, samples (clinical vs. population-based), and analytical methods. Importantly, we highlighted the opportunity for future studies to include covariates that go beyond age and sex, can help refine associations, and can be readily collected. Future studies may want to consider other confounding factors, depending on their research question, design, and assumed causal relations.

## Adjustments for IQ are often unnecessary when examining the relation between brain structure and ADHD

Avoiding bias from adjusting for variables that are not confounders is as important as identifying sources of confounding. Adjusting for mediators or colliders of the ADHD–brain structure relation would induce bias. Here, when adjusting for IQ, which plays an unclear role in brain structure–ADHD associations, cluster sizes reduced considerably in both the ABCD and Generation R Studies. This could indicate that IQ is a confounder, in which case adjustments would be necessary, or that IQ is a mediator or collider, in which case adjustments must be avoided.

First, based on previous literature and this study, the association between ADHD and IQ is relatively weak (*Dennis et al., 2009*) ($r_{ABCD} = -0.11$, $r_{GENR} = -0.14$), but this does not necessarily make it a weak confounder as the strength of confounding is due to a variable's relation with the exposure *and* outcome. Second, if brain structure and ADHD symptoms both cause cognitive changes, adjusting for IQ could induce collider bias, although this is also dependent on when IQ is measured relative to the exposure and outcome (*Hernan and Robins, 2020*). Third, if brain structure determines cognitive

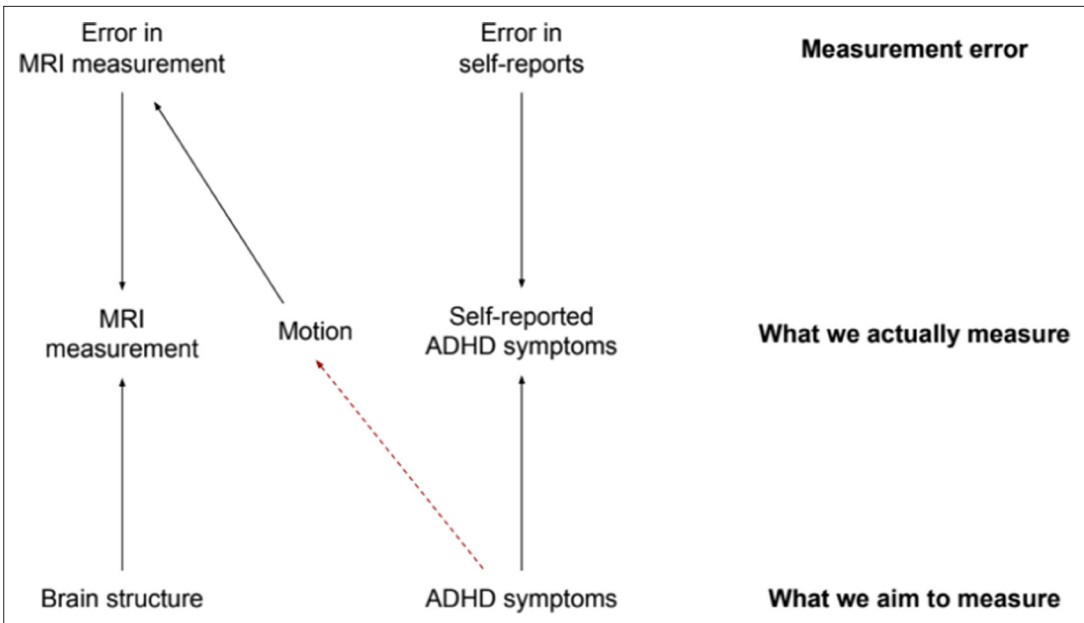

**Figure 7.** Information bias for brain structure, attention-deficit/hyperactivity disorder (ADHD) symptoms, and head motion. *Note.* From the bottom up: We aim to measure the 'true' values of brain structure and ADHD symptoms. However, we actually measure both brain structure and ADHD symptoms imperfectly, at the MRI and through self-reports, respectively. What we assess is therefore affected by measurement errors. Error in the MRI measurement is determined, in part, by excessive motion during scanning. Higher ADHD symptoms likely cause higher motion (dotted red path). This leads to *differential* information bias and creates a non-causal path from ADHD symptoms to brain structure through motion.

functioning, which in turn affects ADHD symptoms (mediation by IQ), adjustments would also induce bias (*VanderWeele, 2016*).

Given these scenarios, we recommend moving away from routinely adjusting for IQ in ADHD neuroimaging studies, and we highlight the need to carefully consider the causal model for a specific research question to determine whether IQ may confound associations.

## There is no easy fix for dealing with head motion in brain morphology–ADHD associations

Adjustments for neuroimaging covariates, such as head motion, are often run to reduce confounding bias. However, head motion, rather than inducing confounding bias, creates measurement error (information bias). When adjusting for head motion during scanning, we observed increases in the spatial extent of the clusters. This might indicate a reduction or an increase in bias. First, bias might have been reduced by adjusting for the fact that children with ADHD will have more error in their cortical measures. Second, bias might have also been increased because we conditioned for head motion, which is a consequence of ADHD.

Overall, the role of head motion in the relation between brain structure and ADHD is complex and warrants the utmost care. Adjusting or not would both lead to bias, meaning that considering which bias might be strongest is necessary. Moreover, methods aiming to reduce information bias should be leveraged (*Lash et al., 2021*); however, further developments are needed for their application to the neuroimaging field.

Similar considerations should be applied to other neuroimaging parameters which often are indicators of information bias rather than confounders in brain structure–ADHD relations (e.g., time of the day, scanner). For instance, time of the day has been shown to influence morphometric values (*Nakamura et al., 2015*; *Trefler et al., 2016*), leading to information bias. If we expect for ADHD symptoms to influence the time of the day in which children with ADHD versus without come to the scanner (differential information bias), the same considerations for head motion would apply. If instead, children with ADHD and without come to the MRI at similar times of the day (non-differential

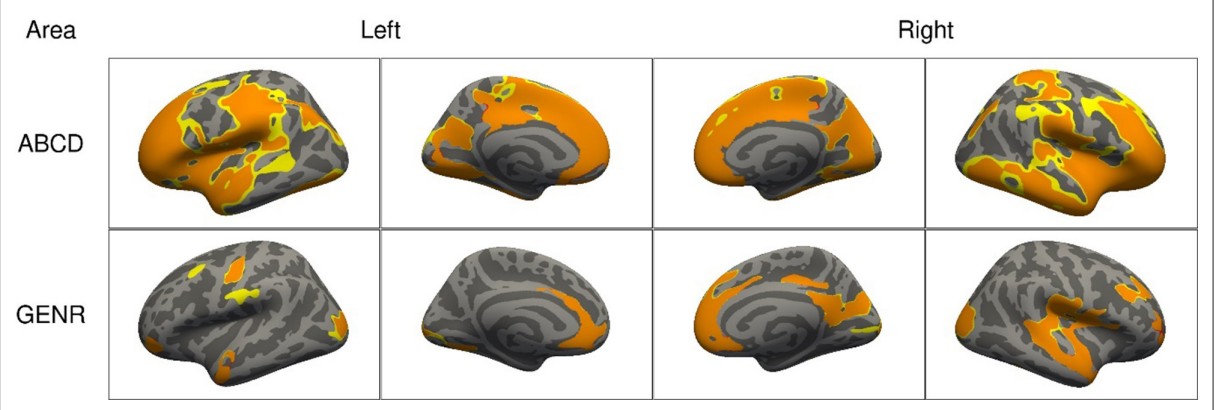

**Figure 8.** Significant clusters in the association of attention-deficit/hyperactivity disorder (ADHD) symptoms with cortical surface area based on the Adolescent Brain Cognitive Development (ABCD) and Generation R Studies, after additional adjustment for motion. *Note.* Rows represent the results for the ABCD or Generation R Studies, and the columns represent the left and right hemispheres. The colors denote the different models, with red vertices being significant only in model 3, orange ones in both model 3 and after adjustment for motion, and yellow ones only after adjusting for motion.

The online version of this article includes the following figure supplement(s) for figure 8:

**Figure supplement 1.** Significant clusters in the association of attention-deficit/hyperactivity disorder (ADHD) symptoms with cortical volume (top) and thickness (bottom) based on the Adolescent Brain Cognitive Development (ABCD) and Generation R Studies, after additional adjustment for motion.

information bias), then adjustments for time of the day would be appropriate and reduce the measurement error.

## Generalization to psychiatric neuroimaging studies

Our considerations on confounding likely generalize to the psychiatric neuroimaging field, as several confounders considered here (e.g., SES) also relate to brain function and other psychiatric disorders (*Biazoli et al., 2020*; *Kivimäki et al., 2020*; *Apter et al., 2017*). Similarly, other psychiatric disorders are also characterized by complex relations with IQ (*Der et al., 2009*). Moreover, mental health problems featuring state anxiety, like internalizing and externalizing symptoms, have also been related to increased head motion during scanning (*Eijlers et al., 2021*).

Confounding control is paramount to studies examining determinants of a phenotype, like ADHD. However, even in these studies, one may be tempted to conduct correlational research with limited confounding adjustments, and then speculate about biological causal mechanisms (*Hernán, 2018*; *Grosz et al., 2020*). Rather, we suggest leveraging prior literature and expert knowledge to identify and adjust for key confounders. This can help eliminate the influence of alternative mechanisms (to the ones hypothesized) on the relation of interest (*Hernan and Robins, 2020*). Charting the assumed (causal) structures to identify confounders can be done through the use of tools such as DAGs (*Hernan and Robins, 2020*). Naturally, the plausibility of such assumptions should be evaluated. To facilitate the minimization of confounding bias in psychiatric neuroimaging, we propose a workflow in *Figure 9*.

## Limitations of the present study and suggestions for future research

Despite leveraging two large samples with similar characteristics and assessments, this study presents several limitations. First, there is always potential for residual confounding through unmeasured confounders and misclassification of measured confounders. For example, given that genetic factors influence both ADHD and brain morphology and that there is a genetic correlation between ADHD risk and intracranial volume (*Jansen et al., 2015*; *Klein et al., 2019*; *Klein et al., 2017*), certain genetic risk variants may be unmeasured confounders. However, we aimed to illustrate plausible confounding bias scenarios for ADHD and brain structure, and not to provide an exhaustive list of potential confounders, which may vary depending on the study population and research question. Future studies should also consider bias analyses to assess the impact that residual confounding may have on the study results (*Lash et al., 2021*). Bias analyses can help understand the minimum association strength an unmeasured confounder needs to have with the determinant and outcome to fully

## Minimizing confounding bias

*A workflow*

### When and Why

Careful confounder adjustments are paramount in studies with an etiological aim. Identify key potential confounders as early as possible and ideally before data collection. Incorporate in your study analysis techniques minimizing confounding bias (e.g. matching, adjustments)

Using models including question- and design-specific confounders minimizes incorrectly classifying effects due to confounding bias as real brain-behavior relations

### Consider multiple DAGs

Where a variable may have an unclear role in the association you are examining, consider multiple DAGs wth different assumptions (e.g. IQ for ADHD and the brain as confounder, collider, mediator). Of note, running multiple models will *not* inform on the role of a variable and none of the assumed models may be correct, but subject-matter knowledge and previous literature can be used to identify likely scenarios.

### Interpretation

Interpret the results in light of the DAG-informed models you previously specified. When interpreting the findings, it is key to evaluate the extent to which you think your DAG model assumptions are met.

### Using and building DAGs

DAGs can be useful to represent research questions and identify confounders, colliders, and mediators. DAGs can therefore guide on which variables should be adjusted for.

Information from previous literature and subject-matter knowledge can be used to guide in building a DAG. Search for information on variables which may affect both the predictor and outcome and make sure to follow DAG rules.

### Consider other sources of bias

Other sources of bias besides confounding may influence your results. Consider the potential biases that may arise from variables not captured in the data (unmeasured confounding), measurement error in confounders (residual confounding), measurement error in the determinant and outcome (information bias), participant selection (selection bias).

**Figure 9.** Suggestions for minimizing confounding bias: a workflow. *Note.* In this workflow, we suggest different actions that can be taken throughout the research process to minimize confounding bias in psychiatric neuroimaging studies.

explain away the findings (*VanderWeele and Ding, 2017*). Developments may be needed, however, for their adaptation to the neuroimaging field.

Second, due to our cross-sectional design, deliberately chosen to correspond to most neuroimaging studies, we must assume all confounders precede our determinant and outcome. This is a plausible assumption for the Generation R Study as, being a prospective birth cohort, we could ensure that the confounders here considered temporally preceded both ADHD and neuroanatomical assessments. However, this was not possible for the ABCD Study, which started sampling at child ages 9–10 years. Future research on the temporal relations between potential confounders, ADHD, and brain structure will aid the minimization of confounding bias when investigating the structural substrates of ADHD.

Third, while we leveraged both symptom-level and diagnostic data for ADHD, this was done within population-based studies. Our results cannot, therefore, be generalized to a clinical population. Future research could examine the extent to which associations between brain structure and ADHD change after adjustments for likely confounders in clinical samples.

In conclusion, leveraging an empirical example from two large studies on neuroanatomy and ADHD symptoms, we highlighted the opportunity for future studies to consider further key confounders. These can be identified based on prior literature and causal diagrams as well as be readily collected, offering a feasible venue for future research. Adjusting for these potential confounders helped identify more refined cortical associations with ADHD symptoms, robust to the influence of demographic and socioeconomic factors, pregnancy exposures, and maternal psychopathology. We also evaluated

the potential role of IQ, which could be a mediator, collider, and/or confounder. While adjusting for IQ led to reductions in associations, these would, however, likely not be attributable to reduced confounding bias. Lastly, we explored how head motion reflects information, rather than confounding bias. We discussed the generalizability of these considerations on confounding bias to psychiatric neuroimaging, and suggest a workflow that can be followed to minimize confounding bias in future studies.

## Materials and methods

### Participants

We analyzed data from two independent population-based cohorts: the ABCD Study and the Generation R Study. The ABCD Study is conducted across 21 study sites in the US and recruited since 2015 children aged 9–10 at baseline (*Garavan et al., 2018*). The Generation R Study is based in Rotterdam, the Netherlands, with data collection spanning from fetal life until early adulthood, and started in 2002 (*Kooijman et al., 2016*). Details of the sampling rationale, recruitment, methods, and procedures have been described elsewhere (*Kooijman et al., 2016*; *Garavan et al., 2018*). Research protocols for the ABCD Study were approved by the institutional review board of the University of California, San Diego (#160091), and the institutional review boards of the 21 data collection sites, while the design of the Generation R Study was approved by the Medical Ethics Committee of the Erasmus MC (METC-2012-165). For both studies, written informed consent and assent from the primary caregiver or child were obtained.

In this cross-sectional study, we leveraged data from the baseline assessment of the ABCD Study (release 2.0.1) and the 10-year assessment of the Generation R Study. Both waves included behavioral and neuroimaging measures. We included children with data on ADHD symptoms and $T_1$-weighted MRI images. Participants were excluded if (1) they had dental braces, (2) incidental findings, (3) their brain scans failed processing or quality assurance procedures, or (4) they were twins or triplets. Of note, excluding children with dental braces is unlikely to determine selection bias by SES in either the ABCD or the Generation R Study as the former cohort covered the costs of dental braces removal

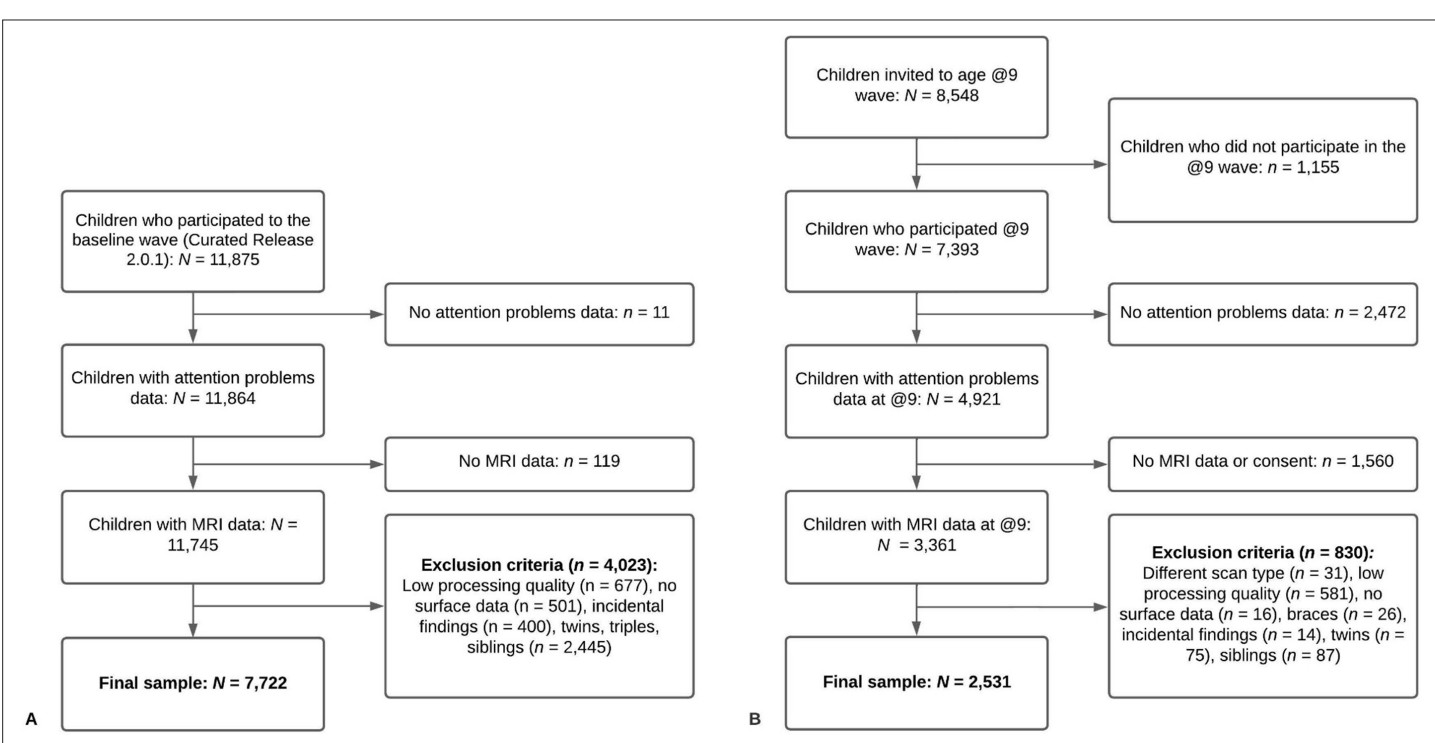

**Figure 10.** Flowcharts of participant inclusion and exclusion for Adolescent Brain Cognitive Development (ABCD) (panel A) and Generation R (panel B). *Note.* (**A**) In the ABCD Study, of the 11,875 participants enrolled in the study, 7722 met our inclusion and exclusion criteria. (**B**) In the Generation R Study, of the 8548 participants invited to the age 9–10 assessment, 2531 met our inclusion and exclusion criteria.

for all children who enrolled, while dental care is insured for all children in the Netherlands. Within the Generation R Study, a small set of participants were additionally excluded because they had a different scan sequence. Finally, for each non-twin sibling set, one was randomly included to minimize shared method variance bias. Flowcharts for participant inclusion and exclusion are available in *Figure 10*. The final samples consisted of 7722 and 2531 children from the ABCD and Generation R Studies, respectively.

## Measures

### ADHD symptoms

Children's ADHD symptoms, reported by the primary caregiver, were measured with the CBCL (school-age version) (*Achenbach, 2001*), an inventory widely used for parent reports of children's emotional and behavioral problems. The attention problem syndrome scale (20 items) measures inattention, hyperactivity, and impulsivity and has been previously shown to have clinical utility and to discriminate between ADHD cases and controls (*Eiraldi et al., 2000*). Attention problems were analyzed on a discrete scale (range 0–19). For the ABCD Study, we repeated the analysis using present ADHD diagnosis from a parent-reported and computerized version of the KSADS-5. This is a dimensional and categorical assessment used to diagnose current and past psychiatric disorders according to the Diagnostic and Statistical Manual of Mental Disorders (Fifth Edition) (*Kaufman et al., 1997*; *Kobak, 2020*).

### Image acquisition

T$_1$-weighted data were obtained on multiple 3T scanners in the ABCD Study (Siemens Prisma, General Electric (GE) 750 and Philips) and one scanner in the Generation R Study (GE 750). Standard adult-sized coils were used for the ABCD Study and an eight-channel receive-only head coil for the Generation R Study. To acquire T$_1$-weighted structural images, the ABCD Study used an inversion prepared RF-spoiled gradient echo scan with prospective motion correction while the Generation R Study used an inversion recovery fast spoiled gradient recalled sequence (GE option = BRAVO, TR = 8.77 ms, TE = 3.4 ms, TI = 600 ms, flip angle = 10°, matrix size = 220 × 220, field of view = 220 × 220 mm, slice thickness = 1 mm, number of slices = 230, ARC acceleration factor = 2). More details can be found elsewhere (*Casey et al., 2018*; *Hagler et al., 2019*; *White et al., 2018*). Of note, in the ABCD Study, a technical mistake occurred at one collection site, causing the hemisphere data to be flipped. This was fixed before processing.

### Image processing

FreeSurfer (version 6.0.0) was used for both cohorts for image processing, which was run in-house to maximize comparability across cohorts. Processing involved (1) removal of non-brain tissue, (2) correction of voxel intensities for B$_1$ field inhomogeneities, (3) tissue segmentation, and (4) cortical surface-based reconstruction. Cortical surface maps were smoothed with a full width of a half-maximum Gaussian kernel of 10 mm. Within the ABCD Study, quality assessment was based on manual and automated quality control procedures and recommended inclusion criteria for structural data from the ABCD team (*Hagler et al., 2019*). Within the Generation R Study, quality assurance was manually performed by visually inspecting all images by trained raters, as previously described in the literature (*Muetzel et al., 2019*). Poor quality reconstructions were excluded.

### Covariate assessment
#### The ABCD Study

All data were collected at baseline (child age 9–10 years). Age and sex were recorded at intake. Child race/ethnicity was reported by the primary caregiver and was categorized as White, Black, Hispanic, Asian, Other by the ABCD team. Household combined net income (<$50,000, ≥$50,000 and <$100,000, ≥$100,000) and highest parental education (<high school, high school diploma/GED, some college, bachelor degree, postgraduate degree) were self-reported by the primary caregiver in the Parent Demographics Survey. Maternal age at childbirth was measured in the Developmental History Questionnaire. Tobacco and cannabis use during pregnancy were retrospectively reported by the mother (yes, no, I do not know) in the Developmental History Questionnaire. Caregiver

psychopathology was obtained from the Total Problems Adult Self Report Syndrome Scale. The Wechsler Intelligence Scale for Children-6 Matrix Reasoning total scaled score was used as a proxy for IQ. We used the Euler number obtained from FreeSurfer as a proxy for head motion during the acquisition. The Euler number quantifies the topological complexity of the reconstructed cortical surface, with holes in the reconstructed surface leading to lower Euler numbers.

## The Generation R Study

Age and sex were measured based on medical records obtained at birth. Child ethnicity (western and non-western) was assessed based on the parents' birth country, in line with the Statistics Netherlands bureau. Maternal age at childbirth was prospectively measured. Family income and highest maternal education were obtained through prospective self-reports by the mother and/or father at child age 5 years. Maternal education was coded into low (no/primary education), intermediate (secondary school, vocational training), and high (Bachelor's degree/University). Household net monthly income was classified as low (<2000 euros), middle (2000–3200 euros), and high (>3200 euros). Maternal postnatal psychopathology, measured at child age 6 months, was prospectively reported by the mother based on the Brief Symptom Inventory questionnaire global severity index. Mothers prospectively reported smoking (never used, used) and cannabis use during pregnancy (no use vs. use during pregnancy). Non-verbal child IQ was measured at child age 5 years, based on the Snijders-Oomen Niet-Verbale Intelligentie Test (*Tellegen and Laros, 1998*), a validated Dutch non-verbal intelligence test. The Euler number was used as a proxy for head motion during scanning.

## Covariate selection

Similar covariates were grouped into confounding sets to minimize the number of tested models while including relevant confounders. Factors included in model 1 related to demographic and study characteristics (age, sex, ethnicity, and study site [for ABCD only]). Age and sex were selected as these have been previously adjusted for in previous neuroimaging studies of ADHD (*Supplementary file 1a*). Ethnicity was used as a proxy for differential health risk exposure among people of different ethnic groups. The study site was incorporated to account for location and scanner differences in the ABCD Study.

Further potential confounders were selected based on previous literature and with the aid of DAGs, as described in the *Results*. In model 2, variables indicating socioeconomic factors were included (parental education, household income, maternal age at childbirth). Household income and parental education are generally considered to measure childhood SES in health research (*American Psychological Association, 2021*). Maternal age at childbirth can additionally inform on the SES of the child by capturing part of the variance unexplained by income and education (e.g., younger mothers facing higher occupational challenges, highly educated mothers delaying childbirth *Heck et al., 1997*). In model 3, maternal factors from the prenatal and postnatal period were grouped (tobacco and cannabis use during pregnancy and maternal psychopathology) to measure early life exposures which may impact a child's brain and psychiatric development.

## Statistical analyses

The R statistical software (version 4.1.0) was used for all analyses. Missing data on covariates were imputed with chained equations using the *mice* R package (*Buuren and Groothuis-Oudshoorn, 2011*). Linear vertex-wise analyses were performed with the *QDECR* R package (*Lamballais and Muetzel, 2021*), with surface area/volume/thickness and ADHD symptoms as variables of interest. Correction for multiple testing was applied by using cluster-wise corrections based on Monte Carlo simulations with a cluster forming threshold of 0.001, which yields false-positive rates similar to full permutation testing (*Greve and Fischl, 2018*). A Bonferroni correction was applied to adjust for analyzing both hemispheres separately (i.e., $p < 0.025$ cluster-wise).

Our analyses for aim 1 involved three vertex-wise linear regression models, which progressively expanded to adjust for additional confounding factors. The first model focused on demographic covariates, the second on socioeconomic ones, and the third on maternal behavioral variables related to psychopathology and pregnancy exposures. These models were run for ADHD symptoms (in both the ABCD and Generation R Studies) and ADHD diagnosis (in the ABCD Study only, as sensitivity analysis). For the results, we primarily reported cluster sizes, that is, the area of the cerebral cortex that

was statistically significantly associated with ADHD symptoms. The sizes of the clusters were reported in $cm^2$.

Two additional models, building upon model 3, were run to address aim 2, to illustrate the consequences of adjusting for factors with a complex and unclear relation with brain structure and ADHD symptoms: IQ and head motion (analyses ran separately). Of note, given that IQ and ADHD were weakly correlated ($r_{ABCD} = -0.11$, $r_{GENR} = -0.14$), multicollinearity was not expected.

## Code availability

Code used to conduct this project is publicly available at https://github.com/LorenzaDA/ADHD_brainmorphology_confounding (Copy archived at swh:1:rev:95c01381fc7fad9bedb3b5918fb80b02b-1dcbdfa) (*Dall'Aglio, 2022*) under CC by 4.0.

## Acknowledgements

The ABCD Study is supported by the National Institutes of Health and additional federal partners under award numbers U01DA041048, U01DA050989, U01DA051016, U01DA041022, U01DA051018, U01DA051037, U01DA050987, U01DA041174, U01DA041106, U01DA041117, U01DA041028, U01DA041134, U01DA050988, U01DA051039, U01DA041156, U01DA041025, U01DA041120, U01DA051038, U01DA041148, U01DA041093, U01DA041089, U24DA041123, U24DA041147. All supporters are mentioned at https://abcdstudy.org/federal-partners.html. Participating sites and study investigators are shown at https://abcdstudy.org/consortium_members/. While ABCD investigators designed, implemented the study, and/or provided data, they did not participate in this manuscript. This work reflects the authors' views, and may not reflect those of the NIH or ABCD investigators. The Generation R Study is supported by Erasmus MC, Erasmus University Rotterdam, the Rotterdam Homecare Foundation, the Municipal Health Service Rotterdam area, the Stichting Trombosedienst & Artsenlaboratorium Rijnmond, the Netherlands Organization for Health Research and Development (ZonMw), and the Ministry of Health, Welfare and Sport. Neuroimaging data acquisition was funded by the European Community's 7th Framework Program (FP7/2008-2013, 212652, Nutrimenthe). Netherlands Organization for Scientific Research (Exacte Wetenschappen) and SURFsara (Cartesius Compute Cluster, https://www.surfsara.nl/) supported the Supercomputing resources. Authors are supported by an NWO-VICI grant (NWO-ZonMW: 016.VICI.170.200 to HT) for HT, LDA, SL, and the Sophia Foundation S18-20, and Erasmus University and Erasmus MC Fellowship for RLM. We thank the participants, general practitioners, hospitals, midwives, and pharmacies in Rotterdam who contributed to the study.

## Additional information

### Funding

| Funder | Grant reference number | Author |
|---|---|---|
| Nederlandse Organisatie voor Wetenschappelijk Onderzoek | NWO-ZonMW: 016.VICI.170.200 | Henning Tiemeier |
| Erasmus Medisch Centrum | Sophia Foundation S18-20 | Ryan L Muetzel |

The funders had no role in study design, data collection, and interpretation, or the decision to submit the work for publication.

### Author contributions

Lorenza Dall'Aglio, Conceptualization, Data curation, Software, Formal analysis, Validation, Investigation, Methodology, Writing – original draft, Project administration, Writing – review and editing; Hannah H Kim, Conceptualization, Data curation, Software, Formal analysis, Validation, Investigation, Methodology, Writing – original draft, Writing – review and editing; Sander Lamballais, Conceptualization, Data curation, Software, Validation, Investigation, Visualization, Methodology, Writing – original draft, Writing – review and editing; Jeremy Labrecque, Conceptualization, Writing – review

and editing; Ryan L Muetzel, Henning Tiemeier, Conceptualization, Resources, Supervision, Funding acquisition, Methodology, Project administration, Writing – review and editing

**Author ORCIDs**
Henning Tiemeier ⓘ http://orcid.org/0000-0002-4395-1397

**Ethics**
Research protocols for the ABCD study were approved by the institutional review board of the University of California, San Diego (#160091), and the institutional review boards of the 21 data collection sites, while the design of the Generation R study was approved by the Medical Ethics Committee of the Erasmus MC (METC-2012-165). For both studies, written informed consent and assent from the primary caregiver or child were obtained.

**Decision letter and Author response**
Decision letter https://doi.org/10.7554/eLife.78002.sa1
Author response https://doi.org/10.7554/eLife.78002.sa2

## Additional files

**Supplementary files**
• Supplementary file 1. Adjustments in prior literature on ADHD and brain structure and descriptive statistics of the present study. (a) Overview of adjusted confounders for neuroimaging studies on attention-deficit/hyperactivity disorder (ADHD). *Note.* S = sensitivity; M = matched. (b) Descriptive statistics of the study population in the ABCD and Generation R Studies. [a]For the ABCD Study, missing values were present for race/ethnicity (<0.1%), highest parental education (<0.1%), household income (8.9%), maternal age at birth (2.4%), smoking during pregnancy (<0.1%), cannabis use during pregnancy (<0.1%), and IQ (2.3%). [b]For Generation R, missing values were present for ethnicity (<0.1%), maternal education (8.0%), household income (12.0%), smoking during pregnancy (11.1%), cannabis use during pregnancy (19.5%), IQ (12.4%), and aggression problems (<0.1%).

• Transparent reporting form
• Reporting standard 1. STROBE checklist.
• Reporting standard 2. STROBE flowchart.

**Data availability**
All datasets for this article are not automatically publicly available due to legal and informed consent restrictions. Reasonable requests to access the datasets should be directed to the Director of the Generation R Study, Vincent Jaddoe (generationr@erasmusmc.nl), in accordance with the local, national, and European Union regulations. Data for The ABCD Study are already open and available in the NIMH Data Archive (NDA) (nda.nih.gov) to eligible researchers within NIH-verified institutions. Data can be accessed following a data request to the NIH data access committee (https://nda.nih.gov/), which should include information on the planned topic of study. The request is valid for 1 year. Data use should be in line with the NDA Data Use Certification. The code used for this study is publicly available at https://github.com/LorenzaDA/ADHD_brainmorphology_confounding (Copy archived at swh:1:rev:95c01381fc7fad9bedb3b5918fb80b02b1dcbdfa).

The following dataset was generated:

| Author(s) | Year | Dataset title | Dataset URL | Database and Identifier |
|---|---|---|---|---|
| Tiemeier H, Muetzel R, Dall'Aglio L, Kim HH, Lamballais S, Labrecque J, Muetzel RL, Tiemeier H | 2022 | Attention-Deficit/ Hyperactivity Disorder symptoms and Brain Morphology: Examining Confounding Bias #1311 | https://doi.org/10.15154/1523058 | NIMH Data Archive, 10.15154/1523058 |

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
