## [Editor Report]

This study provides important and useful information to researchers in brain morphology and ADHD. The strength of the evidence presented is convincing and solid.

---

## [Decision Letter]

**Decision letter after peer review:**

Thank you for submitting your article "Attention Deficit Hyperactivity Disorder Symptoms and Brain Morphology: Examining Confounding Bias" for consideration by *eLife*. Your article has been reviewed by 2 peer reviewers, and the evaluation has been overseen by a Reviewing Editor and Jonathan Roiser as the Senior Editor. The following individual involved in the review of your submission has agreed to reveal their identity: Francisco Xavier Catellanos (Reviewer #1).

Essential revisions:

In this study, the authors examined the associations between brain morphology and ADHD symptoms and how the adjustments for confounders change these associations. The reviewers were generally supportive, but raised several points that should be addressed:

1) The discrepancy between the two datasets needs to be validated/discussed further.

2) Given the nature of ADHD, it is important to evaluate the influence of preprocessing pipeline (quality control steps), motion parameters, and even time of day. The authors should perform the analyses with comparable preprocessing and motion quality between the two datasets to validate the findings.

3) The authors should include cortical thickness on top of the other two measures, and clarify the unit of outcome measures.

*Reviewer #1 (Recommendations for the authors):*

One major concern relates to the choice of outcome measures. The authors focus on vertex-wise surface area and volume and do not include cortical thickness. They note the latter was uninformative in a prior analysis. However, cortical volume is the product of area and thickness, so it's unclear what is gained by including it as if it were an independent measure. In fact, all the maps of differences in volume appear to be reduced versions of the surface area difference maps.

It's also completely unclear why both surface area and volume measures are given in units of cm2. That unit is appropriate for area, but volume is a 3-dimensional measure, so it should be reported in cm3 (or mm3).

These two issues made it difficult to truly address the value of the manuscript as currently submitted. The surface area measures appear to be informative – and they support the authors' contention that potentially confounding variables must be carefully attended to. I am confused and unconvinced regarding the value of the volume.

*Reviewer #2 (Recommendations for the authors):*

Two large datasets were used for replication to strengthen the confidence in the results. However, the significant clusters were different (ABCD showed more widespread regions while GenR showed more localized regions). Is this due to different preprocessing pipelines, motion parameters, or scanner acquisition parameters? Please discuss the possible reasons.

Although the authors have discussed why this study did not consider head motion, it is interesting to see how the scanner acquisition protocol processing parameters and head motion confounds affect the brain morphology-ADHD associations for these two datasets. In addition, from prior literature, time of day also modulates tissue volume metrics (Nakamura et at., 2015, Trefler et al., 2016); therefore it would be interesting to consider time-related confounds as well.

---

## [Author Response]

Essential revisions:In this study, the authors examined the associations between brain morphology and ADHD symptoms and how the adjustments for confounders change these associations. The reviewers were generally supportive, but raised several points that should be addressed:1) The discrepancy between the two datasets needs to be validated/discussed further.

To address this comment, we have now expanded on the discrepancies across the datasets in the Discussion:

Discussion (pages 11-12):

Of note, some inconsistencies between the results from the ABCD and the Generation R studies, both in size of the clusters as well as the exact location, were observed. While we used the same processing pipelines and similar quality control procedures and measures across cohorts, potential reasons for discrepancies of results must be discussed. First, the larger sample size of the ABCD Study allows for greater power to detect smaller effects, which led to larger associated areas. Second, the multi-site structure of the ABCD Study may have introduced noise in the results (e.g., by different scanners, demographic differences), and determined the identification of associations which are not replicable in the Generation R Study. Third, the two studies include children from different populations. While both are very diverse samples, the ABCD Study sample is comprised of a more heterogeneous sample from the U.S. population, which, for instance, is characterized by a wider variety of ethnicities and cultures, potentially permitting the discovery of more associations. Nevertheless, there was considerable overlap in the findings from the ABCD and Generation R Studies, with consistencies across cohorts indicating the most robust and generalizable associations.

2) Given the nature of ADHD, it is important to evaluate the influence of preprocessing pipeline (quality control steps), motion parameters, and even time of day. The authors should perform the analyses with comparable preprocessing and motion quality between the two datasets to validate the findings.

Thank you for this comment. To address it, we further specified the similarities and differences in the pipelines across the two cohorts. First, we clarified the text to better emphasize that many of the processing steps, including the manual quality control, were performed similarly between the two cohorts, with the processing in FreeSurfer being identical. Second, we further aligned the steps in the procedures that we were able to, i.e., applying the inclusion/exclusion criteria in the same order and adding an additional step to exclude incidental findings consistently across the two studies. Third, we mentioned the potential influence of neuroimaging related information, such as head motion and time of the day, in the Discussion section (for in-depth information see Reviewer #2, comment 2).

Changes are shown below:

Methods (page 19):

“FreeSurfer (version 6.0.0) was used for both cohorts for image processing, which was run in-house to maximize comparability across cohorts.”

Methods (page 19):

“Within the ABCD Study, quality assessment was based on manual and automated quality control procedures and recommended inclusion criteria for structural data from the ABCD team (1). Within the Generation R Study, quality assurance was manually performed by visually inspecting all images by trained raters, as previously described in the literature (2).”

3) The authors should include cortical thickness on top of the other two measures, and clarify the unit of outcome measures.

We have now included analyses of cortical thickness. We have also clarified why all findings are reported as cm^2^ (for in-depth information see Reviewer #1, comment 2). Manuscript changes are shown below, in-text and in the Supplement. Overall, few clusters for cortical thickness were related to ADHD symptoms but had similar patterns of reductions with covariate adjustments to clusters for surface area and volume.

Methods (page 21):

“Linear vertex-wise analyses were performed with the QDECR R package (3), with surface area/volume/thickness and ADHD symptoms as variables of interest.”

Methods (page 21):

“For the results we primarily reported cluster sizes, i.e., the area of the cerebral cortex that was statistically significantly associated with ADHD symptoms. The sizes of the clusters were reported in cm^2^.”

Results (page 5):

“We ran vertex-wise linear regression models for ADHD with cortical surface area, volume, and thickness. Results for surface area, which constituted the main findings here, are presented in-text, while findings for volume and thickness in the Supplementary Figures.”

Results (page 5):

“For cortical thickness, we found two small frontal clusters in the ABCD Study (16.1 cm^2^) and no clusters in the Generation R Study, which suggests that cortical thickness does not relate or does not relate strongly to ADHD, in line with prior literature (4) (Figure 1 – Supplement 1).”

Results (page 6):

“Similar reductions were observed for volume and thickness (Figure 3 – Supplement 1).”

Results (page 7):

Compared to clusters for ADHD symptoms, those associated with ADHD diagnosis were smaller, but overlapping (Figure 1 – Supplement 2). We observed similar patterns of reduction in the spatial extent of the clusters after adjusting for each set of confounders (Figure 3 – Supplement 2).”

Results (page 7):

“At a vertex-wise level, adjusting for socioeconomic and maternal factors (model 3) led to reductions in the β coefficients, across the brain, for both cohorts (Figure 4 – Supplement 1). Of note, some β coefficients also showed increases. As confounding bias may lead to under- or over-estimation, it is not surprising to observe both decreases and increases in the average β coefficients after adjustments.”

Results (page 9):

“Findings for volume and thickness are shown in Figure 6 – Supplement 1.”

Results (page 10):

“Results for cortical volume and thickness are shown in Figure 8 – Supplement 1.”

Reviewer #1 (Recommendations for the authors):One major concern relates to the choice of outcome measures. The authors focus on vertex-wise surface area and volume and do not include cortical thickness. They note the latter was uninformative in a prior analysis. However, cortical volume is the product of area and thickness, so it's unclear what is gained by including it as if it were an independent measure. In fact, all the maps of differences in volume appear to be reduced versions of the surface area difference maps.

We thank the Reviewer for this comment. To address it, we have focused on surface area, which, as the Reviewer rightly pointed out, provides the main finding of this project. More specifically, the main text is now built around the results of the analyses of surface area. At the same time, results for volume and thickness (analyses run to address *Essential revisions*, *comment 3*) are also described, partly in-text and in the *Supplementary Figures*.

It's also completely unclear why both surface area and volume measures are given in units of cm2. That unit is appropriate for area, but volume is a 3-dimensional measure, so it should be reported in cm3 (or mm3).

We agree with the Reviewer that this comes across as incorrect. For completeness: The clusters relate to the surface of the cerebral cortex, and the clusters reflect the surface that associates with ADHD symptoms. That surface is expressed in cm^2^. At the level of the vertex we indeed consider different characteristics, such as thickness (mm/cm), area (mm^2^/cm^2^), and volume (mm^3^/cm^3^). The vertices that reach statistical significance cover a certain area, and it is this area that we report in the results. We have clarified the text of the *Methods* section and the *Results section* (to explicitly state it at first occurrence):

Methods (page 21):

“For the results we primarily reported cluster sizes, i.e., the area of the cerebral cortex that was statistically significantly associated with ADHD symptoms. The sizes of the clusters were reported in cm^2^.”

Results (page 5):

“As shown in Figure 1, associations were widespread, as the clusters of association covered 1,165.7 cm^2^ of the cerebral cortex in the ABCD Study, and 446.1 cm^2^ in the Generation R Study.”

These two issues made it difficult to truly address the value of the manuscript as currently submitted. The surface area measures appear to be informative – and they support the authors' contention that potentially confounding variables must be carefully attended to. I am confused and unconvinced regarding the value of the volume.

We hope that we have now sufficiently addressed the Reviewer’s comments and rendered the manuscript more informative.

Reviewer #2 (Recommendations for the authors):Two large datasets were used for replication to strengthen the confidence in the results. However, the significant clusters were different (ABCD showed more widespread regions while GenR showed more localized regions). Is this due to different preprocessing pipelines, motion parameters, or scanner acquisition parameters? Please discuss the possible reasons.

We addressed this important comment by specifying the use of the same processing pipelines and similar motion quality control as well as including a paragraph in the *Discussion section* highlighting potential reasons for inconsistencies across cohorts. Such changes have been aforementioned in Essential Revisions, comment 1. Furthermore, the differences in widespread versus localized findings could reflect greater power in the ABCD Study (due to large sample size) to detect smaller effects. On top of the aforementioned changes (*Essential Revisions, comment 1*), we added a sentence about power as a possible explanation for the differences in the *Discussion section*, as shown below.

Discussion (pages 21-22):

“First, the larger sample size of the ABCD Study allows for greater power to detect smaller effects, which led to larger associated areas.”

Although the authors have discussed why this study did not consider head motion, it is interesting to see how the scanner acquisition protocol processing parameters and head motion confounds affect the brain morphology-ADHD associations for these two datasets. In addition, from prior literature, time of day also modulates tissue volume metrics (Nakamura et at., 2015, Trefler et al., 2016); therefore it would be interesting to consider time-related confounds as well.

We agree with the Reviewer that neuroimaging confounds needed further discussion in our manuscript. We have therefore examined the potential role of head motion in the relations between brain structure and ADHD by using causal diagrams and have additionally adjusted for it in another model. Briefly, head motion, rather than inducing confounding bias, induces measurement error (information bias) in the brain morphology values. Head motion during scanning often results from ADHD symptoms, determining a non-causal path from ADHD to brain structure through motion. In this case, adjustments for head motion can cause both increases or decreases in bias. Increases in bias can occur because head motion is a consequence of ADHD, and consequences of outcomes should not be adjusted for. Decreases in bias can occur because the non-causal path through head motion would be closed with adjustments. Further information is shown in-text and reported below:

Introduction (page 4):

“Of the 19 studies included in a systematic review of neuroimaging studies on ADHD (5), 17 adjusted or matched for age in their analyses, 14 for sex, 9 for neuroimaging-related variables like head motion during scanning, and 8 for the intelligence quotient (IQ)

(Supplementary File 1a).

[…]

Conversely, previous studies have adjusted for IQ and head motion, which may not be confounders in the association between ADHD symptoms and the brain, and may thus have led to further bias in the results (6).

In this study, we examined the association between brain structure and ADHD symptoms and how the selection and control for potential confounders may affect results (aim 1). Moreover, we discussed the complex role of IQ and head motion in brain structure – ADHD associations and the potential consequences of adjusting for them (aim 2).”

Results (pages 9-10):

Head motion does not induce confounding bias, but information bias

A final scenario was also included, to reflect the commonly used adjustments for head motion during scanning (Supplementary File 1a). Motion can be a large source of bias in neuroimaging studies which is important to address. While it does not meet the criteria for confounding as it is not a common cause of ADHD problems and brain morphology (7), head motion can induce measurement error of brain morphology (8) (Figure 7). This is also referred to as information bias and can distort estimates from their true value.

The amount of measurement error in brain morphology may differ across children with vs without ADHD. In fact, children with impulsivity and inattention have been shown to move more during MRI scanning (9,10), determining different levels of error in the brain morphology assessments (Figure 7, path from ADHD symptoms to motion to error in MRI measurement). In this scenario, adjusting for motion might lead to two situations. On one hand, since motion is a consequence of the outcome (ADHD), adjustments would lead to bias (11). On the other hand, not adjusting for motion would also lead to bias because part of the observed relation between ADHD symptoms and brain structure would be due to the higher head motion (and thus the underestimation of the cortical values) of children with ADHD. In this study, we explored the effect of adjusting for motion during scanning in the relation between brain morphology and ADHD (model 5).

Adjustments for motion led to increases in clusters

After additional adjustments for head motion, the spatial extent of the clusters generally increased. For surface area, compared to model 3, clusters increased from 760.2 cm^2^ to 936.4 cm^2^ ( = +23.2%) for the ABCD Study and from 208.6 cm^2^ to 239.7 cm^2^ ( = +14.9%) for the Generation R Study (Figure 8). Clusters of associations consistently found across cohorts were highly similar to the ones identified in model 3. Results for cortical volume and thickness are shown in Figure 8 – Supplement 1.

Discussion (pages 13-14):

There is no easy fix for dealing with head motion in brain morphology – ADHD associations

Adjustments for neuroimaging covariates, such as head motion, are often run to reduce confounding bias. However, head motion, rather than inducing confounding bias, creates measurement error (information bias). When adjusting for head motion during scanning, we observed increases in the spatial extent of the clusters. This might indicate a reduction or an increase in bias. First, bias might have been reduced by adjusting for the fact that children with ADHD will have more error in their cortical measures. Second, bias might have also been increased because we conditioned for head motion, which is a consequence of ADHD.

Overall, the role of head motion in the relation between brain structure and ADHD is complex and warrants the utmost care. Adjusting or not could both lead to bias, meaning that considering which bias might be strongest is necessary. Moreover, methods aiming to reduce information bias should be leveraged (12); however, further developments are needed for their application to the neuroimaging field.

Similar considerations should be applied to other neuroimaging parameters which often are indicators of information bias rather than confounders in brain structure – ADHD relations (e.g., time of the day, scanner). For instance, time of the day has been shown to influence morphometric values (13,14), leading to information bias. If we expect for ADHD symptoms to influence the time of the day in which children with ADHD vs. without come to the scanner (differential information bias), the same considerations as for head motion would apply. If instead, children with ADHD and without will come to the MRI at similar times of the day (non-differential information bias), then adjustments for time of the day would be appropriate and reduce the measurement error.”

Discussion (page 14):

“Our considerations on confounding likely generalize to the psychiatric neuroimaging field, as several confounders considered here (e.g., SES) also relate to brain function and other psychiatric disorders (15–17). Similarly, other psychiatric disorders are also characterized by complex relations with IQ (18). Moreover, mental health problems featuring state anxiety, like internalizing and externalizing symptoms, have also been related to increased head motion during scanning (19).”

Methods (pages 19-20):

For The ABCD Study:

“We used the Euler number obtained from FreeSurfer as a proxy for head motion during the acquisition. The Euler number quantifies the topological complexity of the reconstructed cortical surface, with holes in the reconstructed surface leading to lower Euler numbers.” […]

For The Generation R Study:

“The Euler number was used as a proxy for head motion during scanning.”

References:

1. Hagler DJ, Hatton SeanN, Cornejo MD, Makowski C, Fair DA, Dick AS, et al. (2019): Image processing and analysis methods for the Adolescent Brain Cognitive Development Study. NeuroImage 202: 116091.

2. Muetzel RL, Mulder RH, Lamballais S, Cortes Hidalgo AP, Jansen P, Güroğlu B, et al. (2019): Frequent Bullying Involvement and Brain Morphology in Children. Front Psychiatry 10. https://doi.org/10.3389/fpsyt.2019.00696

3. Lamballais S, Muetzel RL (2021): QDECR: A Flexible, Extensible Vertex-Wise Analysis Framework in R. Front Neuroinformatics 15. https://doi.org/10.3389/fninf.2021.561689

4. Hoogman M, Muetzel R, Guimaraes JP, Shumskaya E, Mennes M, Zwiers MP, et al. (2019): Brain Imaging of the Cortex in ADHD: A Coordinated Analysis of Large-Scale Clinical and Population-Based Samples. Am J Psychiatry 176: 531–542.

5. Saad JF, Griffiths KR, Korgaonkar MS (2020): A Systematic Review of Imaging Studies in the Combined and Inattentive Subtypes of Attention Deficit Hyperactivity Disorder. Front Integr Neurosci 14: 31.

6. Dennis M, Francis DJ, Cirino PT, Schachar R, Barnes MA, Fletcher JM (2009): Why IQ is not a covariate in cognitive studies of neurodevelopmental disorders. J Int Neuropsychol Soc JINS 15: 331–343.

7. Hernan MA, Robins JM (2020): Causal Inference What If? CRC press Taylor and Francis Group.

8. Van de Walle R, Lemahieu I, Achten E (1997): Magnetic resonance imaging and the reduction of motion artifacts: review of the principles. Technol Health Care 5: 419–435.

9. Thomson P, Johnson KA, Malpas CB, Efron D, Sciberras E, Silk TJ (2021): Head Motion During MRI Predicted by out-of-Scanner Sustained Attention Performance in Attention-Deficit/Hyperactivity Disorder. J Atten Disord 25: 1429–1440.

10. Kong X, Zhen Z, Li X, Lu H, Wang R, Liu L, et al. (2014): Individual Differences in Impulsivity Predict Head Motion during Magnetic Resonance Imaging. PLOS ONE 9: e104989.

11. Westreich D (2012): Berkson’s bias, selection bias, and missing data. Epidemiol Camb Mass 23: 159–164.

12. Lash T, Fox M, MacLehose R (2021): Applying Quantitative Bias Analysis to Epidemiologic Data. Springer International Publishing. Retrieved February 16, 2022, from https://link.springer.com/book/9783030826727

13. Nakamura K, Brown RA, Narayanan S, Collins DL, Arnold DL, Alzheimer’s Disease Neuroimaging Initiative (2015): Diurnal fluctuations in brain volume: Statistical analyses of MRI from large populations. NeuroImage 118: 126–132.

14. Trefler A, Sadeghi N, Thomas AG, Pierpaoli C, Baker CI, Thomas C (2016): Impact of time-of-day on brain morphometric measures derived from T1-weighted magnetic resonance imaging. NeuroImage 133: 41–52.

15. Biazoli CE, Salum GA, Gadelha A, Rebello K, Moura LM, Pan PM, et al. (2020): Socioeconomic status in children is associated with spontaneous activity in right superior temporal gyrus. Brain Imaging Behav 14: 961–970.

16. Kivimäki M, Batty GD, Pentti J, Shipley MJ, Sipilä PN, Nyberg ST, et al. (2020): Association between socioeconomic status and the development of mental and physical health conditions in adulthood: a multi-cohort study. Lancet Public Health 5: e140–e149.

17. Apter G, Bobin A, Genet M-C, Gratier M, Devouche E (2017): Update on Mental Health of Infants and Children of Parents Affected With Mental Health Issues. Curr Psychiatry Rep 19: 72.

18. Der G, Batty GD, Deary IJ (2009): The association between IQ in adolescence and a range of health outcomes at 40 in the 1979 US National Longitudinal Study of Youth. Intelligence 37: 573–580.

19. Eijlers R, Blok E, White T, Utens EMWJ, Tiemeier H, Staals LM, et al. (2021, August 11): Internalizing and externalizing behaviors in school-aged children are related to state anxiety during magnetic resonance imaging. medRxiv, p 2021.08.11.21261892.